# Therapeutic Performance Evaluation of ^213^Bi-Labelled Aminopeptidase N (APN/CD13)-Affine NGR-Motif ([^213^Bi]Bi-DOTAGA-cKNGRE) in Experimental Tumour Model: A Treasured Tailor for Oncology

**DOI:** 10.3390/pharmaceutics15020491

**Published:** 2023-02-01

**Authors:** Zita Képes, Viktória Arató, Judit P. Szabó, Barbara Gyuricza, Dániel Szücs, István Hajdu, Anikó Fekete, Frank Bruchertseifer, Dezső Szikra, György Trencsényi

**Affiliations:** 1Division of Nuclear Medicine and Translational Imaging, Department of Medical Imaging, Faculty of Medicine, University of Debrecen, Nagyerdei St. 98, H-4032 Debrecen, Hungary; 2Doctoral School of Pharmaceutical Sciences, University of Debrecen, Nagyerdei St. 98, H-4032 Debrecen, Hungary; 3Doctoral School of Chemistry, Faculty of Science and Technology, University of Debrecen, Egyetem Square 1, H-4032 Debrecen, Hungary; 4Department of Physical Chemistry, Faculty of Science and Technology, University of Debrecen, Egyetem Square 1, H-4032 Debrecen, Hungary; 5European Commission, Joint Research Centre (JRC), 76344 Karlsruhe, Germany

**Keywords:** APN/CD13 (Aminopeptidase N), [^213^Bi]Bi-DOTAGA-cKNGRE, fibrosarcoma, HT1080, NGR (asparagine-glycine-arginine), positron emission tomography (PET), tumour volume

## Abstract

Since NGR-tripeptides (asparagine-glycine-arginine) selectively target neoangiogenesis-associated Aminopeptidase N (APN/CD13) on cancer cells, we aimed to evaluate the in vivo tumour targeting capability of radiolabelled, NGR-containing, ANP/CD13-selective [^213^Bi]Bi-DOTAGA-cKNGRE in CD13pos. HT1080 fibrosarcoma-bearing severe combined immunodeficient CB17 mice. 10 ± 1 days after cancer cell inoculation, positron emission tomography (PET) was performed applying [^68^Ga]Ga-DOTAGA-cKNGRE for tumour verification. On the 7th, 8th, 10th and 12th days the treated group of tumourous mice were intraperitoneally administered with 4.68 ± 0.10 MBq [^213^Bi]Bi-DOTAGA-cKNGRE, while the untreated tumour-bearing animals received 150 μL saline solution. In addition to body weight (BW) and tumour volume measurements, ex vivo biodistribution studies were conducted 30 and 90 min postinjection (pi.). The following quantitative standardised uptake values (SUV) confirmed the detectability of the HT1080 tumours: SUV_mean_ and SUV_max_: 0.37 ± 0.09 and 0.86 ± 0.14, respectively. Although no significant difference (*p* ≤ 0.05) was encountered between the BW of the treated and untreated mice, their tumour volumes measured on the 9th, 10th and 12th days differed significantly (*p* ≤ 0.01). Relatively higher [^213^Bi]Bi-DOTAGA-cKNGRE accumulation of the HT1080 neoplasms (%ID/g: 0.80 ± 0.16) compared with the other organs at 90 min time point yields better tumour-to-background ratios. Therefore, the therapeutic application of APN/CD13-affine [^213^Bi]Bi-DOTAGA- cKNGRE seems to be promising in receptor-positive fibrosarcoma treatment.

## 1. Introduction

Angiogenesis/neo-angiogenesis has a central role in tumour development, tumour progression and metastatic spread [1,2]. Several biomarkers of tumour-associated angiogenesis are known from the literature, which can be targeted by different peptides (for example: vascular endothelial growth factor (VEGF) derivatives for vascular endothelial growth factor receptor (VEGFR), RGD for integrins and NGR for APN/CD13). Although NGR tripeptide sequence is currently the least investigated angiogenesis-connected agent, the ever-increasing importance of APN/CD13 in association with cancer-related processes, diagnostics or as a prognostic biomarker and therapeutic target brings NGR motif and its derivatives into the priority of today’s science [3]. Former findings indicate that membrane-bound metalloprotease, aminopeptidase N (CD13/APN) takes part in the differentiation, movement, invasion, proliferation and apoptosis of the cells, as well as angiogenic processes [4]. In addition, tumour vascular endothelial cells and various cancer cells including prostate, pulmonary, gynaecological, breast and gastrointestinal ones could be characterised by CD13 upregulation [5,6,7,8,9,10]. Since neo-angiogenesis-associated exopeptidase CD13/APN is widely expressed on different neoplasms and angiogenic endothelial cells, it serves as a highly reliable diagnostic as well as prognostic biomarker of tumour-related angiogenic processes [1,11]. Therefore, CD13 selective imaging as well as therapeutic probes could be ground-breaking in early-stage tumour identification and personalised treatment of CD13pos. malignancies [12].

Ample evidence implies that asparagine–glycine–arginine sequence (tripeptide NGR)-containing peptides exhibit high affinity to CD13/APN [13]. Hence, NGR motif-based peptide molecules could be essential in the targeted imaging of CD13 overexpressing neoplastic alterations. In this respect, nuclear medicine has a lot to offer. Prior literature data proved that radiolabelled NGR-based molecular agents make a leap forward in the non-invasive diagnostic assessment of cancers with high APN density [14,15]. Given the broad accessibility of different radiometals for NGR radiolabelling purposes such as copper-64 (^64^Cu), gallium-68 (^68^Ga), lutetium-177 (^177^Lu) or rhenium-188 (^188^Re) together with the high-quality and non-invasive imaging modalities, a rising number of research studies have been centred around the evaluation of APN-directed radio-appended peptide compounds in the recent years [16,17,18,19].

Previous in vivo investigations outlined the diagnostic feasibility of the following ^64^Cu or ^68^Ga radio-conjugated probes in receptor pos. experimental tumour models at translational level: ^64^Cu-labelled NGR monomer and dimer attached to chelator DOTA (1,4,7,10-tetraazadodecane-N,N′,N″,N′′′-tetraacetic acid; [^64^Cu]Cu-DOTA-NGR and [^64^Cu]Cu-DOTA-NGR_2_), ^68^Ga-labelled cyclic NGR (cNGR) linked to NOTA (1,4,7-triazacyclononane-triacetic acid; [^68^Ga]Ga-NOTA-cNGR) or NODAGA (1,4,7-triazacyclononane-1-glutaric acid-4,7-diacetic acid; [^68^Ga]Ga-NODAGA-cNGR) [17,20,21]. Out of the broad set of chelators—including NOTA, NODAGA, DOTA, or 1,4,7,10-tetrakis(carboxymethyl)-1,4,7,10-tetraazacyclododecane glutaric acid (DOTAGA)—due to its high stability and the inertness of the complexes formed with either ^68^Ga and ^213^Bi radiometals, DOTA is recognised as the most promising macrocyclic chelator for diagnostic and therapeutic applications. To facilitate complexation with targeting peptides, several bifunctional derivatives of DOTA were developed, such as DOTAGA, by using an additional carboxy functionality added to the core of DOTA [22,23].

Although the past few decades are characterized by outstanding advancements in battling against tumourigenesis, cancer-associated death remains one of the major challenges of current healthcare systems worldwide [24]. To overcome the limitations of conventional anti-tumour treatment, targeted radionuclide therapy (TRNT) using alpha-emitting peptide-specific molecules represents a pioneering therapeutic approach in oncological fields [25,26]. The high linear energy transfer (LET: 50–230 keV/µm) and the short penetration range (0.1 mm) of alpha particles are well-suited for selective tumour-killing, bypassing adverse effects on neighbouring healthy tissues [27,28]. Produced by actinium-225/bismuth-213 (^225^Ac/^213^Bi) generator system, alpha-emitting ^213^Bi (T_½_: 45.6 min, mean E_α_/MeV: 8.32, average soft tissue range: 78 µm) with such valuable physical properties seems to be a potent weapon in TRNT-based anti-tumour treatment [29,30].

In the present study we aimed at evaluating the therapeutic performance of a newly synthesised, [^213^Bi]Bi-labelled, NGR-motif containing radiopharmaceutical in CD13/APN-upregulated HT1080 fibrosarcoma-bearing mice. Prior to its therapeutic administration, positron emission tomography (PET) with ^68^Ga-labelled CD13-targeting compound was applied for tumour verification.

## 2. Materials and Methods

(2,2′,2″-(10-(1-Carboxy-4-((4-Isothiocyanatobenzyl)Amino)-4-Oxobutyl)-1,4,7,10-Tetraazacyclododecane-1,4,7-triyl)Triacetic Acid). p-NCS-Bn-DOTAGA was purchased from Chematech (Dijon, France) and cKNGRE from CASLO ApS (Lyngby, Denmark). Ultra-purified (u.p.) hydrochloric acid, NaOAc and water were purchased from Merck KGaA (Darmstadt, Germany). ^68^Ga radioisotope was eluted with 0.1 M u.p. HCl from a germanium-68/gallium-68 (^68^Ge/^68^Ga) isotope generator (Eckert-Ziegler, GalliaPharm^®^, Berlin, Germany). Activity measurements were carried out with a CAPINTEC CRC-15PET dose calibrator and a Perkin Elmer Packard Cobra gamma counter (Llantrisant, UK). Oasis HLB 1cc cartridge was purchased from Waters Corporation (Milford, MA, USA). All other reagents were purchased from Sigma-Aldrich.

### 2.1. Radiolabelling

#### 2.1.1. Radiolabelling of [^68^Ga]Ga-DOTAGA-cKNGRE

The NGR-motif-containing peptide was produced according to Gyuricza et al. [31]. The chemical structure of the DOTAGA-cKNGRE is presented in Figure 1.

Briefly, in the presence of triethylamine (0.046 mmol), the NH_2_ group of the cKNGRE (0.011 mmol) was reacted with p-NCS-Bn-DOTAGA (0.01 mmol) in a mixture of dimethylformamide and water (3:1, 1 mL). After stirring for one day the mixture was diluted with water and lyophilised. The crude DOTAGA-cKNGRE was then purified by semi-preparative HPLC using a KNAUER HPLC system with a semi-preparative Supelco Discovery^®^ Bio Wide Pore C18 10 µm (150 × 10 mm) column, connected to a UV detector. Eluents were A: 0.1% trifluoroacetic acid (TFA) in water and B: 0.1% TFA in 95% ACN.

The radiolabelling of DOTAGA-cKNGRE with ^68^Ga was accomplished according to Gyuricza et al. [31]. This method is based on formerly published labelling procedures [32,33,34]. One mL (100–130 MBq) ^68^GaCl_3_, eluted with 0.1 M u.p. HCl from a ^68^Ge/^68^Ga generator, was mixed with sodium acetate buffer (0.16 mL, 1 M) to obtain pH4. Then, 18 nmol of DOTAGA-cKNGRE was added to the solution and finally the mixture was incubated for 5 min at 95 °C. Then, [^68^Ga]Ga-DOTAGA-cKNGRE was purified by solid-phase extraction (SPE) applying an Oasis HLB 1cc cartridge. Analitical radio-HPLC was applied for the quality control (QC) of the purified, ^68^Ga-labelled complex. Analytical radio-HPLC was conducted using a KNAUER HPLC system, connected to an ATOMKI CsI scintillation detector (eluent A: 0.1% TFA in water and eluent B: 0.1% TFA in 95% ACN). The radiochemical purity (RCP) of the [^68^Ga]Ga-DOTAGA-cKNGRE was higher than 95%.

#### 2.1.2. Radiolabelling of [^213^Bi]Bi-DOTAGA-cKNGRE

For radiolabelling, ^213^Bi]BiI_4_^−^/BiI_5_^2−^ ion was eluted from ^225^Ac/^213^Bi generator (180 MBq, Institute for Transuranium Elements (ITU)) using 0.6 mL of a mixture of 0.1 M HCl and 0.1 M NaI (1:1). Then the [^213^Bi]BiI_4_^−^/BiI_5_^2−^ eluate was added to a vial containing 135 μL 2 M TRIS buffer, 50 μL 20% ascorbic acid and 30 μL of 6 mM DOTAGA-cKNGRE solution. The reaction mixture was incubated at 95 °C for 15 min. The ^213^Bi-labelled complex was purified with SPE using Oasis HLB 1 cc cartridge pre-conditioned with 5 mL EtOH and 10 mL water. After loading the reaction mixture, the cartridge was rinsed with water (1 mL). The ^213^Bi-labelled complex was eluted with a mixture of water and ethanol (1:1, 250 µL) and diluted with physiological saline (Salsol, TEVA, Debrecen, Hungary). For quality control, the RCP of the purified radiotracer was determined by applying instant thin-layer chromatography (iTLC) by measuring the activity of the 440 keV gamma emission of ^213^Bi using miniGITA TLC scanner. For radio-iTLC, glass microfiber chromatography paper impregnated with a silica gel (iTLC-SG paper) as a stationary phase and 0.5 M citric acid (pH 5.5) as an eluent were used. Free ^213^Bi moves with the solvent front, while [^213^Bi]Bi-DOTAGA-cKNGRE remains at the bottom of the iTLC paper. RCP of the [^213^Bi]Bi-DOTAGA-cKNGRE was found to be above 99% and the molar activity was 0.5 MBq/nmol.

### 2.2. LogP Measurement

#### 2.2.1. Determination of LogP Value of [^68^Ga]Ga-DOTAGA-cKNGRE

The distribution of the radioactivity in *n*-octanol as well as water was determined by Gyuricza et al. in order to define the *LogP* value of [^68^Ga]Ga-DOTAGA-cKNGRE [31]. To have the two different layers separated, the mixture of 50 µL from the given radiotracer (approximately 2 MBq), 450 µL distilled water and 0.5 mL *n*-octanol was centrifuged for 5 min (9000× *g*). Then, the radioactivity was measured in 20 and 20 µL aliquots of *n*-octanol and water, respectively, with a Perkin Elmer Packard Cobra gamma-counter for the determination of the partition coefficient of the investigated probe. The *LogP* value was −4.13 for [^68^Ga]Ga-DOTAGA-cKNGRE.

#### 2.2.2. Determination of LogP Value of [^205/206^Bi]Bi-DOTAGA-cKNGRE

For the determination of *LogP* value we used ^205/206^Bi isotopes as surrogates which were produced by GE PETtrace cyclotron using the method described by Lagunas-Solar et al. [35]. The radiolabelling of the precursor was carried out as follows: 50 µL of 3 M NH_4_OAc buffer (pH 4) and 15 µL of 6 mM DOTAGA-cKNGRE aqueous solution were added to 50 µL [^205/206^Bi]BiCl_3_ in 0.1 M HCl. The labelled complex was purified as described above. Then 10 µL of the purified [^205/206^Bi]Bi-DOTAGA-cKNGRE in physiological saline (Salsol, TEVA, Debrecen, Hungary) was diluted with 0.49 mL water. It was then mixed with 0.5 mL *n*-octanol in a vial and shaken for 5 min. After centrifugation (6000 rpm, 5 min) the radioactivity of the sample from both the aqueous and *n*-octanol phase was measured with a Perkin Elmer Packard Cobra gamma counter. The *LogP* value was −2.59.

### 2.3. Stability Studies

#### 2.3.1. Serum Stability of [^68^Ga]Ga-DOTAGA-cKNGRE

Following the addition of 50 μL of [^68^Ga]Ga-DOTAGA-cKNGRE dissolved in physiological saline (Salsol, TEVA, Debrecen, Hungary) to 50 μL of mouse plasma, the mixture was incubated at 37 °C. With the application of radio-HPLC we evaluated the serum stability of the radiotracer 0, 30, 60 and 90 min post-incubation with mouse blood serum under predefined circumstances. [^68^Ga]Ga-DOTAGA-cKNGRE demonstrated metabolic stability in serum after 90 min since the sample taken from the mixture showed an RCP above 95%, which meets the requirements for ^68^Ga-labelled radiopharmaceuticals.

#### 2.3.2. Serum Stability of [^205/206^Bi]Bi-DOTAGA-cKNGRE

Fifty μL of [^205/206^Bi]Bi-DOTAGA-Bn-cKNGRE in physiological saline (Salsol, TEVA, Debrecen, Hungary) was added to the solution of fifty μL of mouse plasma. The RCP of the samples was determined by radio-iTLC at 0, 30, 60 and 90 min time points using the same method as described above. After 90 min the [^205/206^Bi]Bi-DOTAGA-Bn-cKNGRE remained intact as the RCP of the sample taken from the mixture was above 95%, which meets the requirements for ^213^Bi-labelled radiopharmaceuticals.

### 2.4. Cell Culturing

HT1080 (human fibrosarcoma) cell line was obtained from American Type Culture Collection (ATCC, Manassas, VA, USA). Cell culture was grown in GlutaMAX™ Dulbecco’s Modified Eagle’s Medium (DMEM) (Gibco™, Beijing, China) supplemented with 1% antimycotic (*v*/*v*) (Gibco™), and 1% antibiotic solution (*v*/*v*) (Gibco™) and 10% heat-inactivated fetal bovine serum (*v*/*v*) (FBS, Gibco™). Using T75 flasks (Sarstedt Ltd., Budapest, Hungary) the tumour cells were cultured at 37 °C in ESCO CCL-170B-8 dedicated cell culture incubator characterised by the following parameters: 5% CO_2_ atmosphere and 95% humidity. The transplantation of the HT1080 fibrosarcoma cells was accomplished after five cell passages. For the identification of viability, we employed trypan blue exclusion test.

### 2.5. Animal Housing

Twelve-week-old old severe combined immunodeficient CB17 (SCID) female mice (n = 35) were purchased from Charles River Laboratories (Sulzfeld, Germany). The experimental animals were bred and obtained under specific pathogen-free conditions in Individually Ventilated Cages (Sealsafe Blue line IVC system, Techniplast, Akronom Ltd., Budapest, Hungary) with regulated ambient temperature and relative humidity maintained at 24 ± 2 °C and 55 ± 10%, respectively. A circadian light/dark cycle of 12 h was provided for the study mice. All animals were kept on sterile semi-synthetic rodent maintenance chow (Akronom Ltd., Budapest, Hungary) and sterile drinking water ad libitum.

All procedures applied followed the applicable sections of the Hungarian Laws as well as the animal welfare directions and regulations of the European Union. For the accomplishment of the present study, the permission of the Ethics Committee for Animal Experimentation of the University of Debrecen, Hungary (ethical permission number: 16/2022/DEMÁB) was granted. The fulfilment of the 3R policy was our priority.

### 2.6. HT1080 Tumour Generation

Experimental tumour induction was carried out under inhalation anaesthesia with a dedicated small animal anaesthesia device (Tec3 Isoflurane Vaporizer, Eickemeyer Veterinary Equipment, Luton, UK) applying 3% isoflurane (Forane, AbbVie, Chicago, IL, USA), 0.4 L/min O_2_ and 1.4 L/min N_2_O. This was followed by depilation and disinfection subcutaneous (s.c.) inoculation of 1.5 × 10^6^ HT1080 (human fibrosarcoma) cells in 100 μL (1/3 part of Matrigel and 2/3 part of saline) into the left shoulder area of the CB17 SCID mice. For tumour verification the study mice underwent in vivo [^68^Ga]Ga-DOTAGA-cKNGRE-based PET imaging 10 ± 1 days post tumour cell inoculation. To evaluate the organ distribution pattern of the ^213^Bi-labelled therapeutic peptide compounds, ex vivo biodistribution studies were also executed 10 ± 1 days following the HT1080 cell transplantation at an average tumour volume of 155 ± 20 mm^3^. Thereafter, tumour sizes based on the largest and the smallest tumour diameters were registered ((largest diameter × smallest diameter^2^)/2). We used calliper measurements for tumour growth determination.

### 2.7. In Vivo MiniPET Imaging

CB17 SCID mice bearing a HT1080 fibrosarcoma in their left shoulder area were intravenously (i.v.) injected with 9.36 ± 1.24 MBq [^68^Ga]Ga-DOTAGA-cKNGRE 10 ± 1 days after tumour generation at a tumour bulk of approximately 150 mm^3^. MiniPET scans were acquired 90 min post radiotracer administration with the application of a MiniPET scanner (University of Debrecen, Faculty of Medicine, Division of Nuclear Medicine and Translational Imaging). Isoflurane-based inhalation anaesthesia (AbbVie, Budapest, Hungary; OGYI-T-1414/01) was assured throughout the whole study employing the Tec3 Isoflurane Vaporizer small animal anaesthesia device (Eickemeyer Veterinary Equipment, Luton, UK). Three-dimensional ordered-subsets expectation-maximization line-of-response (3D OSEM-LOR) image reconstruction was performed to provide satisfactory PET images for interpretation. To determine the following uptake parameters, volume of interests (VOIs) were deposited around selected organs and tissues with the BrainCAD image analysis software: standardised uptake value (SUV), SUV_mean_ and SUV_max_. Calculated by the following formula, SUV was used as a relative measure of radiopharmaceutical accumulation: SUV = [VOI activity (Bq/mL)]/[injected activity (Bq)/animal weight (g)]. The average radioactivity within the VOI is reflected by the SUV_mean_ value, while SUV_max_ refers to the highest radiotracer concentration of the region concerned.

### 2.8. Cancer Treatment Studies

Seven days after HT1080 cell inoculation the tumour-bearing animals were randomly classified into two subgroups: treatment-naïve group (control, n = 8) and [^213^Bi]Bi-DOTAGA-cKNGRE-treated group (treated, n = 8). Approximately 5 MBq [^213^Bi]Bi-DOTAGA-cKNGRE was intraperitoneally (i.p.) administered to the experimental mice of the treated cohort on the 7th, 8th, 10th and 12th experimental days. The control mice were i.p. injected with 150 μL saline solution applying the same time regime. 

### 2.9. Ex Vivo Biodistribution Studies

To assess the organ distribution of [^213^Bi]Bi-DOTAGA-cKNGRE, ex vivo investigations were performed. The HT1080 fibrosarcoma-bearing experimental mice were anaesthetised with Forane (5%) and were sacrificed at 30 and 90 min post administration of the radiopharmaceutical. After autopsy, selected tissues, organs and the tumour itself were removed, weighed wet and their radioactivity was measured on a calibrated gamma counter (Perkin-Elmer Packard Cobra, Waltham, MA, USA). Then, the activity was decay corrected to the time of the injection. The percent of administered dose per gram of tissue (%ID/g) was determined for all samples from the counts per minute values (CPM). We presented the ex vivo data as mean %ID/g ± SD.

### 2.10. Statistical Analyses

Statistical analyses were executed with the application of MedCalc 18.5 commercial software package (MedCalc 18.5, MedCalc Software, Mariakerke, Belgium). Statistical differences were determined applying two-tailed Student’s *t*-test, two-way ANOVA and Mann–Whitney U-test. Data are shown as the mean ± SD. A *p* value of less than 0.05 was assumed to identify significant differences unless otherwise indicated.

## 3. Results and Discussion

### 3.1. [^68^Ga]Ga-DOTAGA-cKNGRE MiniPET Imaging

In a bid to authenticate the presence of the HT1080 fibrosarcoma tumours developing in the left shoulder area of the study mice, in vivo whole body MiniPET images were acquired 90 min pi. of [^68^Ga]Ga-DOTAGA-cKNGRE 10 ± 1 days after s.c. tumour induction. Within the framework of the in vivo studies, the tumour targeting potential of the CD13-affine ^68^Ga-tagged peptide probe was also assessed. Figure 2A displays representative in vivo decay-corrected transaxial and coronal PET images of CB17 SCID mice bearing HT1080 tumour. SUV_mean_ and SUV_max_—as quantitative uptake parameters—were registered as well (as seen in Figure 2B).

Upon qualitative assessment the s.c. progressing HT1080 tumours were clearly identified with the application of [^68^Ga]Ga-DOTAGA-cKNGRE 90 min pi. and 10 ± 1 days following tumour cell implantation (presented in Figure 2A, red arrows). This considerable tumour uptake together with the discrete background activity led to the acquisition of high-contrast PET images. In accordance with the visual interpretation, quantitative SUV analyses also revealed meaningful radiopharmaceutical uptake in the neoplastic tissue with SUV_mean_ and SUV_max_ values of 0.37 ± 0.09 and 0.86 ± 0.14, respectively (as shown in Figure 2B). Taken together, our in vivo PET results confirm that the investigated ^68^Ga-appended, cKNGRE-motif-containing imaging probe accumulates in the APN/CD13 expressing tumour.

Accordingly, the tumour-specificity of dimeric cNGR radiolabelled with ^68^Ga and conjugated to chelator DOTA (^68^Ga-DOTA-cNGR_2_) was strengthened in CD13 overexpressing ES2 ovarian cancer bearing nude mice by in vivo microPET imaging [36]. Added to this, reduced ES2 ^68^Ga-DOTA-c(NGR)_2_ tracer uptake induced by the co-application of the cold peptide derivate also outlined the APN-selectivity of the assessed peptide probe. In a similar way, Szabó et al. further confirmed the diagnostic applicability of ^68^Ga-tagged-c(NGR) linked to NOTA in the detection of subrenally growing primary mesoblastic nephrome (Ne/De) tumours (SUV_mean_: 4.12 ± 0.56; SUV_max_: 10.72 ± 1.85), and related metastases in the thoracic parathymic lymph nodes (PTLN; SUV_mean_: 0.72 ± 0.12; SUV_max_: 1.92 ± 0.58) in Fischer-344 rats [37]. In accordance with our results, high-contrast PET images of the PTLNs could be acquired with [^68^Ga]Ga-NOTA-c(NGR) [37]. Obtaining scans with appropriate tumour-to-background ratios still represents a fundamental challenge in terms of image reporting since sharp distinction of the neoplastic tissue from the background activity is of paramount importance regarding precise lesion detection and localization. Corresponding to the findings of Szabó et al., in the experiments of Máté and co-workers, a high affinity of [^68^Ga]Ga-NOTA-c(NGR) to ANP/CD13 pos. ortho and heterotopic transplanted Ne/De tumours was observed [16].

Given the aforementioned in vivo research findings, ^68^Ga-labelled NGR sequence-containing radiotracers have a promising role in the non-invasive detection of cancer-related neo-angiogenic processes as well as the timely identification of APN/CD13-rich tumours. Further, they could be also feasible in the follow-up of angiogenesis directed oncological treatments and the evaluation of therapeutic efficacy.

### 3.2. Performance Evaluation of [^213^Bi]Bi-DOTAGA-cKNGRE Treatment: Effects on Body Weight and Tumour Volume

On the 7th, 8th, 10th and 12th days of the investigation at an average tumour volume of 26.56 ± 2.39 mm^3^ CB17 SCID mice bearing HT1080 fibrosarcoma in their left shoulder area were i.p. injected with approximately 5 MBq [^213^Bi]Bi-DOTAGA- cKNGRE. The treatment-naïve study mice i.p. received 150 μL saline solution on the same experimental days. In a bid to assess the therapeutic efficacy of the investigated probe, body weights (BW, expressed in grams/g ± SD) and tumour volume (mm^3^ ± SD) of the small animals were registered. Calliper measurements were applied to evaluate tumour development.

#### 3.2.1. Impact of [^213^Bi]Bi-DOTAGA-cKNGRE Treatment on Body Weight

Rigorous measurements of BW took place on the 7th, 8th, 10th and 12th experimental days. As demonstrated on the follow-up curve of Figure 3A, no statistically significant change was pinpointed regarding the BW of the experimental small animals in either study group during the therapy (*p* ≤ 0.05). The BW in the untreated cohort was 19.84 ± 1.24, 19.99 ± 1.35, 18.82 ± 1.11, 18.31 ± 0.97 recorded on the 7th, 8th, 10th and 12th days, respectively. As for the [^213^Bi]Bi-DOTAGA- cKNGRE-treated subclass the following values were obtained 7, 8, 10 and 12 days after the tumour cell inoculation: 20.12 ± 1.04, 20.93 ± 0.85, 19.98 ± 0.97 and 19.11 ± 1.07, respectively. Since the weight of the tumourous animals remained relatively stable throughout the whole examination period, we may presuppose that the currently applied [^213^Bi]Bi-DOTAGA-cKNGRE treatment exerts no considerable effect either on tumour weight or BW. Hence, this alpha-emitting NGR-based anti-tumour molecule may constitute a novel, safe therapeutic alternative in the existing treatment avenue of fibrosarcoma. As the evolution of BW is dependent upon a myriad of factors; however, additional long-term studies are warranted to confirm our hypothesis.

#### 3.2.2. Impact of [^213^Bi]Bi-DOTAGA-cKNGRE Treatment on Tumour Volume

Tumour volumes of both small animal subgroups were identified and recorded on the 7th, 8th, 9th, 10th and 12th days of the investigation. The follow-up curves representing tumour progression are presented in Figure 3B; a steady expansion of the HT1080 tumours was encountered until the termination of the research with the tumour volumes of 204.43 ± 28.36 and 112.50 ± 18.42 mm^3^ measured on the 12th day in the untreated control and the [^213^Bi]Bi-DOTAGA-cKNGRE treated subgroups, respectively. We determined the subsequent tumour volumes in the treatment-naïve cohort: 24.88 ± 3.15 mm^3^ (on day 7), 40.67 ± 5.98 mm^3^ (on day 8), 63.00 ± 11.53 mm^3^ (on day 9), 155.43 ± 20.47 mm^3^ (on day 10) and 204.43 ± 28.36 mm^3^ (on day 12), whereas as for the treatment-administered subcategory tumour bulks of 28.25 ± 2.67, 23.33 ± 6.87, 35.36 ± 7.57, 58.33 ± 14.36 and 112.50 ± 18.42 mm^3^ measured on the 7th, 8th, 9th, 10th and 12th days, respectively represented continuous size incrementing.

A statistically meaningful disparity was observed between the tumour volumes of the untreated and the treated mice on the 9th, 10th and 12th days post-tumour cell implantation (*p* ≤ 0.01). Albeit, no remarkable disproportion was observed between the tumour progression of the two groups on the remaining experimental days (*p* ≤ 0.05, on day 7 and 8). As no significant alterations of tumour bulk were assessed during the administration of the first two therapies, we may draw the conclusion that at least two treatment regimes are necessary to induce measurable changes regarding tumour growth. Given the relatively permanent weight of the tumourous animals during the investigation and the experienced remarkable difference between the tumour volume of the treated and the untreated mice on the 9th, 10th, and 12th experimental days, we may draw the conclusion that the reduction of the tumour mass was not as significant in terms of affecting the BW of the study animals. In addition, the fact that the consistency of the tumours might become more compact may underpin why the meaningful tumour size reduction did not impact tumour weight. Taking into account the above remarked findings, application of [^213^Bi]Bi-DOTAGA-cKNGRE appears to be an efficient tool in targeted fibrosarcoma treatment.

Although no previous research has been published so far on the administration of NGR-containing radioactive molecules in the treatment of fibrosarcoma, Maggiorella et al. dealt with the treatment of HT1080 tumours at preclinical level [38]. In their study, hafnium oxide-containing NBTXR3 nanopartices (NPs) were designed to broaden the therapeutic window of radiotherapy—delivered with a cobalt-60 source—in nude NMRI mice bearing HT1080 tumours. Notable enhancement (mean dose enhancement factor at 4 and 8 Gy above 1.5) in the radiation response of HT1080 tumourous mice was observed applying irradiation-activated NPs. Further no clonogenic toxicity associated with the intratumourally injected NPs was remarked in the tumour-bearing study mice.

To the best of our knowledge, this is the first study to evaluate the anti-cancer competence of radiolabelled, APN/CD13-affine NGR-based molecules. Peptide-selective therapeutic probes appended with alpha-emitting ^213^Bi ensure directed tumour cytotoxicity sparing the surrounding intact tissue. Lack of unwanted side effects of the healthy organs together with no treatment-related BW reduction emphasise the safety of the labelled derivates. However, their future progression from bench-to-bedside could be hampered by some constraints around the usage of ^213^Bi. Given the short half-life of the radiometal, problems with radiotracer transfer may arise. In addition, peptide selection must also be adjusted to the physiological half-life of ^213^Bi. A constructive logistic approach is warranted to overcome these limitations to exploit the beneficial therapeutic effects of ^213^Bi.

### 3.3. Ex Vivo Organ Distribution Studies

To assess the organ distribution pattern of the [^213^Bi]Bi-DOTAGA-cKNGRE, ex vivo studies were conducted 30 and 90 min p.i. of approximately 5 MBq [^213^Bi]Bi-DOTAGA-cKNGRE. The ex vivo data are displayed in Figure 4 and Appendix A. After dissection, the following organs and tissues were harvested, and measured for radioactivity by γ-counting: blood, liver, spleen, kidney, small and large intestines, stomach, muscle, fat, lung, heart, brain, bone (femur), salivary glands, gall bladder, pancreas and HT1080 tumour. Except for the liver, the radiotracer accumulation of the examined organs and tissues and the tumour itself decreased gradually over the experimental period (from 30-to-90 min). The highest uptake of the assessed probe—registered in the kidney at both time points—underpinned its primary role in the excretion of the radiolabelled NGR-based molecule (11.36 ± 1.41, 4.51 ± 1.21, 30- and 90 min pi.; respectively). In line with our observations, ex vivo evaluation of ^68^Ga-tagged CD13-affine peptide derivate (^68^Ga-NOTA-Gly_3_-CNGRC/^68^Ga-NOTA-G_3_-NGR) also confirmed outstanding renal accumulation in a study evaluating the tumour-targeting potential of the mentioned probe in HT1080 fibrosarcoma with high APN density and CD13-negative HT-29 human colon adenocarcinoma-bearing mice an hour after the administration of 370 kBq of [^68^Ga]Ga-NOTA-G_3_-NGR [39]. Hence, it seems that neither the change of the chelator (from DOTAGA to NOTA) nor the radiometal affects renal uptake.

Moreover, notably reduced ^213^Bi-probe accumulation could be depicted in the blood (2.72 ± 0.22 and 0.38 ± 0.33; 30 and 90 min p.i., respectively), the kidneys (11.36 ± 1.41, 4.51 ± 1.21; 30 and 90 min p.i., respectively), the small (0.69 ± 0.20; 0.16 ± 0.08; 30 and 90 min p.i., respectively) and the large intestines (0.71 ± 0.06, 0.26 ± 0.08; 30 and 90 min p.i., respectively), the muscle (0.49 ± 0.09, 0.06 ± 0.02; 30 and 90 min p.i., respectively) and the pancreas (0.57 ± 0.18, 0.08 ± 0.03; 30 and 90 min p.i., respectively) between the two experimental time points (*p* ≤ 0.01). The blood (2.72 ± 0.22), the lung (2.09 ± 0.74) and the s.c. developing HT1080 tumours (1.48 ± 0.18) were identified with moderate/relatively high radioactivity 30 min post-tracer administration. The muscle, the bone (femur), the gall bladder and the pancreas exhibited discrete tracer accumulation with ID%/g figures of 0.49 ± 0.09, 0.69 ± 0.38, 0.49 ± 0.31 and 0.57 ± 0.18 30 min after [^213^Bi]Bi-DOTAGA-cKNGRE injection, respectively and 0.06 ± 0.02, 0.08 ± 0.05, 0.17 ± 0.24 and 0.08 ± 0.03 90-min uptake values, respectively. Similarly, modest muscular and osseous [^68^Ga]Ga-NOTA-G_3_-NGR tracer accumulation was recorded in a former biodistribution study conducted by Shao and co-workers [39].

Relatively moderate tracer uptake registered at 30 min p.i. in the intestines showed a statistically significant decline until the termination of the ex vivo studies (*p* ≤ 0.01). The observed gastrointestinal (GIT) activity could be attributed to the physiological CD13 expression of the intestinal epithelial cells [40,41]. This uptake kinetics of the digestive organs may indicate the existence of a GIT way of elimination beyond renal clearance. We hypothesise that the rising radioactivity of the liver (0.72 ± 0.35 and 0.89 ± 0.64; 30 and 90 min after the injection, respectively) may be partially explained by reticuloendothelial cell (RES)-mediated radiotracer uptake and subsequent prolonged radiopharmaceutical retention in the parenchyma of the liver. Although no existing research findings support our assumption, we presume that the molecular size and the physicochemical properties of the investigated probes induce RES-directed tracer accumulation. We further suggest that hepatic vascularisation and factors associated with the metabolism of the liver could also underpin the reason behind the enhanced hepatic uptake. Albeit, long-term, unbiased future studies are warranted to strengthen our hypotheses. Moreover, excretion through the hepatobiliary system could also be supposed based on the elevated liver tracer accumulation. The statistically considerable drop of blood radioisotope uptake from 2.72 ± 0.22 to 0.38 ± 0.33 within 60 min refers to the rapid blood clearance of the investigated probe (*p* < 0.01). In a similar way, prior research study evaluating the ex vivo uptake kinetics of ^68^Ga-appended NGR-containing radioligands—[^68^Ga]Ga-NODAGA-NGR and [^68^Ga]Ga-NOTA-(NGR)_2_—also encountered fast elimination from the blood pool [42].

In our study, the radioactivity was almost entirely cleared from the muscle within 1.5 h. We suppose that factors related to the vasculature could probably undermine the rapid clearance from the muscle tissue. Nevertheless, the experienced statistically (*p* < 0.01) remarkable decrease of the faint pancreatic radiopharmaceutical accumulation over the examination period could be connected to the tracer metabolism. In a like manner, former ex vivo research comparing the pancreatic uptake of [^64^Cu]Cu-NOTA-RGD-NGR between a KCH genetically engineered mouse model bearing integrin α_V_β_3_ and CD13pos. pancreatic ductal adenocarcinoma (PDAC) and healthy control counterparts with normal pancreas revealed insignificant radioactivity in the tumour naïve pancreas of the control cohort [1]. Our organ distribution analyses showed the second highest [^213^Bi]Bi-DOTAGA-cKNGRE accumulation in the lungs. Relatively high activities of the lungs with uptake figures of 2.09 ± 0.74 and 0.71 ± 0.44 measured 30 and 90 min p.i., respectively, refer to the pulmonary retention of the labelled probe. Based on the research findings of Hajdu et al. we may suppose that the prolonged dissemination of the hydrophilic [^213^Bi]Bi-DOTAGA-cKNGRE in pulmonary chambers with water content may underlie this notable tracer uptake [43]. The entrance of the probe to the central nervous system could be hampered by the blood–brain barrier causing insignificant brain activity.

The tumour-targeting potential of [^213^Bi]Bi-DOTAGA-cKNGRE was also evaluated in HT1080 tumour-bearing CB17 mice. The labelled derivate accumulated rapidly in the tumours producing images of adequate equality 1.5 h post-administration that confirmed the APN/CD13 affinity of the radio-conjugated derivative. Although the tumourous tracer activity decreased from 30 to 90 min post-administration, relatively higher accumulation of the HT1080 neoplasms compared with the other organs at 90 min time point yields better tumour-to-background (T/M) ratios and image contrast. The tumour-to-organs ratio is of particular concern in terms of diagnostics since more contrasted images ensure precise lesion identification as well as localisation. In accordance with our results, Shao et al. also observed tumour-specific prompt accumulation (4.96 ± 3.18 ID%/g) of CD13-selective [^68^Ga]Ga-NOTA-G_3_-NGR with a comparable contrast level to that of ours investigating CD13pos. HT1080 fibrosarcoma-bearing nude mice with static microPET imaging [39]. Moreover, other studies also noted the diagnostic efficacy of NGR-based peptides radiolabelled with ^68^Ga in well-differentiated SMMC 7721-based APNpos. hepatocellular carcinoma (HCC) and CD13 upregulated HT1080-bearing nude BALB/c mice (%ID/g: 2.17 ± 0.21 and 2.46 ± 0.23 for the HCC and the HT1080 tumours, respectively, (*p =* 0.18 > 0.05) [44]. Additional investigations also strengthened the imaging potential/diagnostic potential of ^68^Ga-labelled NGR sequence-based PET probes linked to different chelators including DOTA, DOTAGA, NODAGA and N, N′-bis[2-hydroxy-5-(carboxyethyl)benzyl]ethylenediamine-N,N′-diacetic acid (HBED-CC) in CD13pos. A549 (human lung small cell carcinoma) and HT1080 preclinical tumour models [45,46,47]. Based on these results we can conclude that these novel imaging agents selectively target APN regardless of the type of the chelator.

Since the overexpression of pro-angiogenic cell surface molecules such as CD13 on neoplastic as well as tumour vascular endothelial cells has a central role in cancer-related neo-angiogenesis, radiolabelled APN peptide-targeted ligands could represent a landmark in directed anti-tumour treatment [16]. Given the favourable accumulation profile of [^213^Bi]Bi-DOTAGA-cKNGRE in HT1080 oncological models, it would be a promising therapeutic drug candidate for incorporation into standard-of-care fibrosarcoma treatment protocols. Beyond therapeutic purposes, concomitant gamma radiation of ^213^Bi together with satisfactory image quality make the radiometal applicable for diagnostic usage as well.

## 4. Conclusions

Given the adequate binding competence of [^68^Ga]Ga-DOTAGA-cKNGRE to APN/CD13-overexpressing neoplasms, this PET imaging probe serves as a silver bullet for the precise identification of receptor pos. primary malignancies as well as pertinent metastases. Our results fuel the view that alpha-emitting ^213^Bi-tagged, APN-targeting NGR-motif—[^213^Bi]Bi-DOTAGA-cKNGRE—represents a breakthrough in current oncological therapies, leading to the ultimate goal of the establishment of bespoke, directed theranostic cancer treatment in the foreseeable future.

## Figures and Tables

**Figure 1 pharmaceutics-15-00491-f001:**
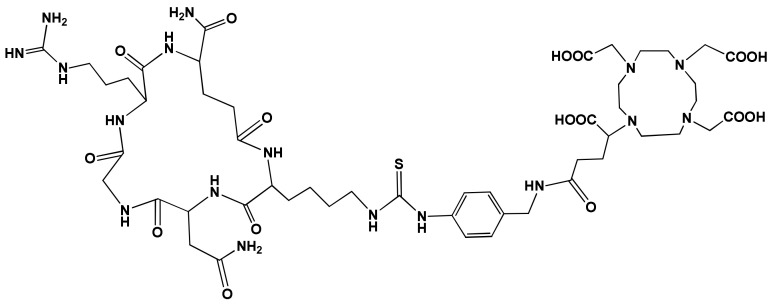
Chemical structure of DOTAGA-cKNGRE.

**Figure 2 pharmaceutics-15-00491-f002:**
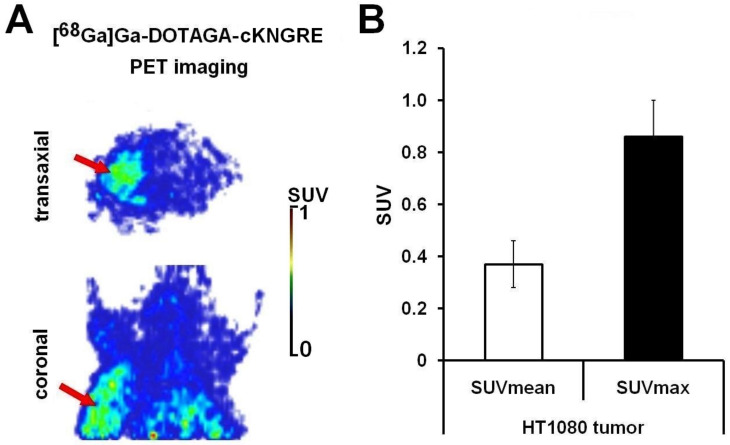
In vivo assessment of ^68^Ga-labelled DOTAGA-conjugated cKNGRE ([^68^Ga]Ga-DOTAGA-cKNGRE) uptake in HT1080 tumour-bearing CB17 SCID mice. (**A**) demonstrates the representative decay-corrected transaxial and coronal PET images of the tumourous study mice 90 min after the intravenous injection of [^68^Ga]Ga-DOTAGA-cKNGRE, 10 ± 1 days after subcutaneous tumour generation (**A**). (**B**) represents the quantitative SUV analyses of the radiotracer uptake of the HT1080 tumours (n = 5) 90 min post tracer administration and 10 ± 1 days after the subcutaneous injection of HT1080 tumour cells. SUV data are presented as mean ± SD. Red arrows: HT1080 tumour. ^68^Ga: gallium-68; DOTAGA: 1,4,7,10-tetrakis(carboxymethyl)-1,4,7,10-tetraazacyclododecane glutaric acid; cKNGRE: cyclic Lysine–Asparagine–Glycine–Arginine–Glutamic acid; SCID: severe combined immunodeficient; PET: positron emission tomography; SUV: standardised uptake value; SD: standard deviation.

**Figure 3 pharmaceutics-15-00491-f003:**
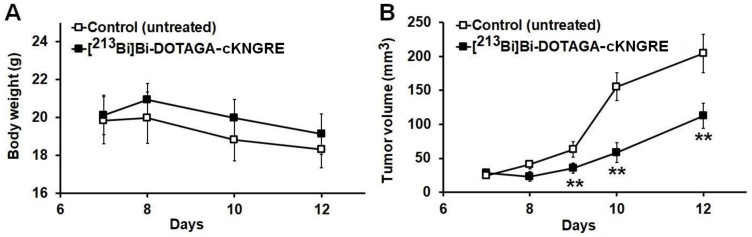
Impact of [^213^Bi]Bi-DOTAGA-cKNGRE treatment on the body weight (grams/g) (**A**) and the tumour growth (mm^3^) (**B**) changes of HT1080 tumour-bearing experimental mice. Treatment administration began 7 days after the subcutaneous injection of HT1080 fibrosarcoma cancer cells at an average tumour volume of approximately 25 mm^3^. Significance level between the untreated control group (n = 5) and the [^213^Bi]Bi-DOTAGA-cKNGRE treated group (n = 5) was *p ≤* 0.01 (**) (**B**). ^213^Bi: bismuth-213; DOTAGA: 1,4,7,10-tetrakis(carboxymethyl)-1,4,7,10-tetraazacyclododecane glutaric acid; cKNGRE: cyclic Lysine–Asparagine–Glycine–Arginine–Glutamic acid.

**Figure 4 pharmaceutics-15-00491-f004:**
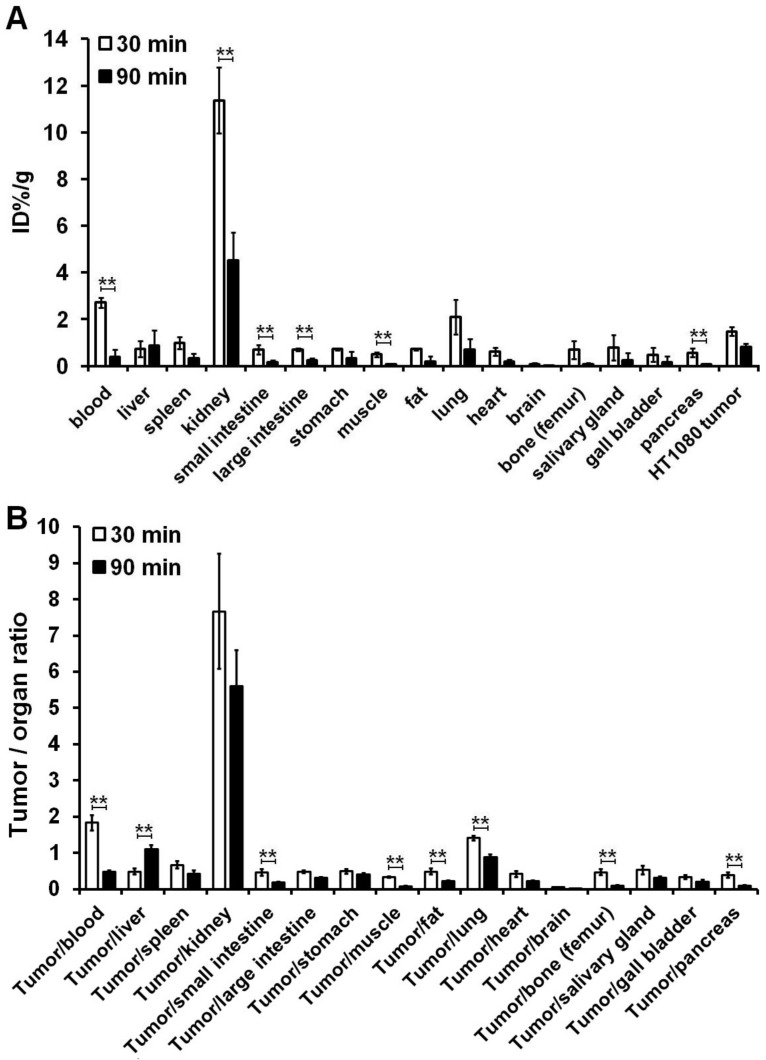
Ex vivo biodistribution (**A**) of [^213^Bi]Bi-DOTAGA-cKNGRE and tumour-to-organ ratios (**B**) in HT1080 tumour-bearing study mice 30 and 90 min postinjection of approximately 5 MBq of the APN/CD13 selective radiotracer. The uptake values were obtained as mean %ID/g ± SD. n = 8 mice/time point. Significance level between 30 and 90 min for the corresponding radiopharmaceutical: *p* ≤ 0.01 (**). ^213^Bi: bismuth-213; DOTAGA: 1,4,7,10-tetrakis(carboxymethyl)-1,4,7,10-tetraazacyclododecane glutaric acid; cKNGRE: cyclic Lysine–Asparagine–Glycine–Arginine–Glutamic acid; APN: aminopeptidase N; SD: standard deviation.

## Data Availability

The datasets used and/or analysed during the current study are available from the corresponding author upon reasonable request.

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
