# Peer review of "Therapeutic Performance Evaluation of 213Bi-Labelled Aminopeptidase N (APN/CD13)-Affine NGR-Motif ([213Bi]Bi-DOTAGA-cKNGRE) in Experimental Tumour Model: A Treasured Tailor for Oncology"

_pharmaceutics, 2023, doi:10.3390/pharmaceutics15020491_

Round 1

Reviewer 1 Report

The authors describe studies in a xenograft model of Ga-68 and Bi-213 DOTAGA complexes of a cyclic NGR peptide for imaging and therapy using neoangiogenesis as a target. Ga-68 was able to image the xenograft tumours and Bi-213 showed indications of growth delay.

The manuscript is very thorough in most aspects.

Although the introduction is fairly detailed, two aspects of this work are not discussed: why the authors chose NGR rather than the more widely studied RGD, and why they chose DOTAGA as the chelator over others which have been used, even by these authors. It would be useful for the reader to understand the rationale for these decisions.

I have a number of minor points.

Page 3, section 2.1.2., para 1. The first sentence states: “DOTAGA-cKNGRE was prepared as described by Gyuricza et al. [31].” I have two questions. The first paragraph of the previous section 2.1.1. described preparation of DOTAGA-cKNGRE. Also, ref 31 (previous work by the same group) used RGD rather than NGR targeting. Please clarify.

Page 3, section 2.1.2., para 2. Please state which ITLC medium was used. I believe there are two available: silica gel and silicic acid

Page 3-4, LogP determination. Small point, but why were two different centrifuge speeds used? Different laboratories? Secondly, the LogP for the Bi complex was stated but not for the Ga-68 complex.

Page 7, last sentence. I don’t agree that the PET studies “confirm the APN/CD13 selectivity” of the imaging probe. They demonstrate accumulation in the tumour but not selectivity.

Page 8, para 2, line 2. I think it is premature to say that NGR based radiotracers have an “undoubtable” role in detection of neoangiogenesis. Perhaps “promising”.

Page 9, first full para, lines 6-8. This result needs to be worded more clearly. I don’t understand why the authors state that the treatment does not affect tumour weight. Figure 3 panel B suggests a considerable effect on tumour size, in terms of growth delay, as discussed in the next section.

Page 10, Section 3.3., para 2. There are a lot of numbers quoted in this paragraph. They might be presented more clearly in a table.

Page 13, Conclusion, line 1. I think that “outstanding binding competence” is too strong a description.

TYPOS ETC

Page 1, Abstract, line 2. Suggest replacing “at evaluating” with “to evaluate”

Page 1, Abstract, line 5. Suggest replacing “executed” with “carried out” or “performed”

Page 1, Abstract, line 7. Units of Bi-213 activity missing

Page 1. Abstract, line 8. Suggest replacing “Besides” with “In addition to”

Page 1, Abstract, line 12. Suggest changing “disproportion” to “difference”

Page 2, Introduction, para 1, line 1. “metastatisation” is not an English word; should be replaced with “metastasis” or “metastatic spread”

Page 2, Introduction, para 2, line 2. Suggest changing “seem to be” to “could be”

Page 3, Section 2.1.2, line 1. Remove “labelled” because Figure 1 shows the peptide, not the labelled peptide. The figure caption is correct.

Page 3, Section 2.1.2, line 2. Suggest replacing “demonstrated” with “presented”

Page 3, last line. Replace “centrifugated” with “centrifuged”

Page 5, Section 2.7., line 3. “3” should be superscript

Page 8, Section 3.2., line 2. Here it states that the administered activity was 8 MBq, while on page 5, Section 2.8., line 3, it says 5 MBq. Which is correct?

Page 11, 2nd full para, line 2. Suggest replacing “derivate” with “derivative”

The references are not formatted consistently

The manuscript will require some editing for English grammar and idiom

Reviewer 2 Report

I appreciated the results of this novel theranostic approach to CD13/APN expressing tumors.  The following are not criticisms but rather suggestions to further improve your paper

* Results of QC and stability of the Bi-complex are worthy of being deepened, with regard to possible impurities which can influence normal tissue dosimetry.

*   Liver is the only organ where accumulation increases from 30 to 90 min pi.  This aspect may be relevant from the dosimetric point of view if Bi-complex could be employed in the clinical practice.  If RES is involved one should hypothesize the presence of colloidal impurities, but spleen activity tends to decrease rendering this hypothesis less acceptable. A deeper discussion may be valuable

* A further figure depicting Target to Non-Target ratios 3o and 90 min p.i. (as described in the text) is worth to be included

* 3.3 ex vivo : “30, - and 90 minutes pi”  why not “30 and 90 minutes pi”

* pag 8, 5th row,  pag. 10, last row  :      “in a similar vein”    (way ?)
